# A Comparative Study of Clinical and Molecular Features of Microsatellite Stable Colorectal Cancer With and Without Liver Metastases

**DOI:** 10.3390/cancers17223677

**Published:** 2025-11-17

**Authors:** Tara Magge, Svea Cheng, Shuaichao Wang, Masood Pasha Syed, Bhaghyasree Jambunathan, Ashley Mcfarquhar, Paola Zinser Peniche, Doga Kahramangil Baytar, Aatur Singhi, Anwaar Saeed, Ibrahim Halil Sahin

**Affiliations:** 1University of Pittsburgh School of Medicine, Pittsburgh, PA 15213, USA; 2Department of Internal Medicine, University of Pittsburgh Medical Center, Pittsburgh, PA 15213, USA; 3University of Pittsburgh Medical Center Hillman Cancer Center, Pittsburgh, PA 15232, USA; 4Department of Pathology, University of Pittsburgh School of Medicine, Pittsburgh, PA 15213, USA; 5Department of Medicine, Division of Hematology & Oncology, University of Michigan Medical School, Ann Arbor, MI 48109, USA

**Keywords:** colorectal cancer, liver metastasis, treatment resistance, chemotherapy, immunotherapy and biomarkers

## Abstract

Liver metastasis of colorectal cancer (CRC) is associated with worse survival outcomes and inferior response to immunotherapy. However, the molecular underpinnings of this phenomenon are not well established. Our study revealed that the incidence of molecular alterations is relatively similar between liver metastases and non-liver metastases of CRC, indicating the treatment resistance seen with liver metastasis is likely driven by intrinsic tumor microenvironment characteristics of the liver. In our study, we also identified that liver metastasis of CRC is associated with shorter time on frontline therapy, a surrogate for treatment response. This suggests that inferior treatment response seen with liver metastasis is not limited to immunotherapy but also may apply to chemotherapy. We also discovered the metastatic site-specific impact of driver oncogenes, such as BRAF and KRAS mutations, pointing out that the impact of driver alterations on survival outcomes can vary depending on the site of metastasis of CRC.

## 1. Introduction

In the United States, colorectal cancer (CRC) is the third most common cancer in both men and women and the second most common cause of cancer mortality, with an estimated 150,000 new cases and 53,000 deaths in 2020 [1]. The majority of CRC-related deaths can be attributed to metastatic disease, which has a five-year survival rate of less than 20% [2]. Metastatic disease to the liver is particularly common among patients with CRC, with approximately 70% of patients with advanced CRC having liver involvement [3,4,5,6].

The mainstay of treatment for unresectable metastatic CRC is systemic therapy, which includes chemotherapy, targeted therapy, immunotherapy, and their combinations [7,8]. Immune checkpoint inhibitors (ICIs) are highly effective agents in treating patients with microsatellite instability-high (MSI-H)/mismatch repair-deficient (dMMR) CRC, which represent 5% of advanced CRC [9,10,11], but they demonstrated relatively limited activity in microsatellite-stable (MSS)/mismatch repair-proficient (pMMR) CRC [3,12]. Recent studies suggest that the presence of liver metastasis in advanced CRC may also be associated with resistance to ICI therapy, independent of mismatch repair status [13,14,15,16,17]. A recent study of nivolumab and regorafenib in advanced CRC demonstrated an overall response rate of 21.7% in patients with non-liver metastasis compared to 0% in those with liver metastasis [18]. The presence of liver metastasis is also associated with poor prognosis for patients with advanced colorectal cancer [19], including shorter median overall survival, lower disease control rate, and shorter progression-free survival on ICIs [17,18,20,21]. Further research revealed that the presence of liver metastasis may confer resistance to other immune-modulating agents beyond ICIs, such as innate/adaptive immune activators [22].

The mechanism of treatment resistance leading to worse clinical outcomes in patients with liver metastasis is not well-defined. Furthermore, while resistance to immunotherapy has been established, it remains unclear if resistance also pertains to chemotherapy and other systemic treatments. Therefore, in this study, we performed comparative analysis on the molecular characteristics of advanced-stage CRC with liver metastasis versus non-liver metastasis and investigated the impact of liver metastasis on both overall survival and response to systemic therapy among patients with advanced-stage MSS CRC. We then characterized the metastatic site-specific impact of common molecular alterations on overall survival and response to treatment.

## 2. Methods

### 2.1. Study Population and Data Collection

Patients diagnosed with metastatic microsatellite stable (MSS) CRC from 2014 to 2022 were identified using the institutional molecular CRC database of the University of Pittsburgh School of Medicine and University of Pittsburgh Medical Center, and its Network. Eligible patients were those diagnosed with colorectal cancer with available next-generation sequencing (NGS) results from genomic profiling. Demographic and clinical data, including but not limited to age at diagnosis, sex, race/ethnicity, site of primary tumor, and site of metastasis at the presentation of metastatic disease, were collected by reviewing electronic medical records. Molecular data of interest, including alterations in KRAS, NRAS, TP53, PIK3CA, BRCA2, and ERBB2 (HER2), were abstracted from the molecular database. Microsatellite instability (MSI) status and tumor mutational burden (TMB) were abstracted using the Oncomine-based NGS database. HER2 amplification was determined from copy number data. TMB subclassification cut-off was determined to be 10 mutations/Mb. The study was conducted under the University of Pittsburgh IRB-approved protocol (STUDY20070085).

### 2.2. Statistical Analysis

Survival analyses focused on individuals with metastatic colorectal cancer. Overall survival (OS) was defined as the time from metastatic disease to death from any cause. Time to next treatment was defined as the time between first-line and second-line systemic treatment. Patients who lost their clinical follow-up were censored according to their last contact date. Kaplan–Meier curves were used to estimate OS, and the log-rank test was used for group comparisons. Multivariable Cox proportional hazards regression was used to evaluate associations between OS and clinical variables, including site of metastasis and molecular alterations. Hazard ratios (HRs), 95% confidence intervals (CIs), and two-sided *p*-values were reported accordingly. Comparisons between groups were performed using the chi-square or Fisher’s exact test for categorical variables and the Mann–Whitney U test for continuous variables.

### 2.3. Molecular Testing

We abstracted molecular data from an in-house molecular database, which utilizes Oncomine Comprehensive Assay v3 (Oncomine) testing, in which targeted NGS-based testing from DNA and mRNA are analyzed in the MGP lab at UPMC using the Oncomine Comprehensive Assay v3 (Oncomine) DNA and RNA primer sets (Thermo Fisher Scientific in Pittsburgh, PA, USA) by using the manufacturer’s protocol. Further details regarding molecular testing can be found in our other publications [23].

## 3. Results

Among a cohort of 299 patients with advanced CRC, the median age was 62.9 years, 42% (126/299) were female, and 58% (173/299) were male. 78% of patients (233/299) had cancer of the colon and 21% (64/299) had cancer of the rectum. Among patients with advanced CRC, 69% (205/299) had liver metastasis, while 31% (94/299) had non-liver metastasis (Table 1). Liver metastasis was slightly more common in male patients with advanced-stage CRC (70%, 121/173) than female patients (67%, 84/126), though not statistically significant (*p* = 0.614). Liver metastasis occurred at significantly higher rates in patients with colon cancer (74%, 173/233) compared to patients with rectal cancer (48%, 31/64) (*p* = 0.00013) (Table 2).

We identified that 49% of patients (147/299) had KRAS mutations, 3.7% (11/299) had NRAS mutations, 6.7% (20/299) had BRAF mutations, 81% (241/299) had TP53 mutations, 4.7% (14/299) had BRCA2 mutations, 15% (44/299) had PIK3CA mutations, and 1% (3/299) had ERBB2 amplification. Overall, the tumor mutational burden (TMB) was 8.86 mutations/Mb, with 38% of patients (115/299) with greater than ten mutations/Mb (Table 2). There was no significant difference in the incidence of KRAS, NRAS, BRAF, or other molecular alterations tested, including TP53, BRCA2, and PIK3CA, in patients with liver metastasis compared to patients with non-liver metastasis (Table 2). There was also no significant difference in TMB when patients with liver and non-liver metastases were compared. Among patients with KRAS mutations, there was no significant difference in the incidence of exon 2 and non-exon 2 KRAS mutations in the liver metastasis cohort compared to the non-liver metastasis cohort (Table 2).

The median overall survival was 43.1 months for the overall population. Although not statistically different, patients with liver metastasis had a numerically worse median overall survival of 42.2 months compared to patients with non-liver metastasis, who had a median overall survival of 47.6 months (*p* = 0.27) (Figure 1A). Multivariate Cox regression analysis revealed that among molecular alterations, both KRAS mutations and BRAF mutations significantly predicted worse overall survival in patients with advanced CRC (hazard ratio 2.12, *p* < 0.001, and hazard ratio 3.37, *p* < 0.001, respectively), consistent with the literature (Table 3). We then examined the potential impact of molecular alterations in patients with liver metastasis specifically, and the presence of KRAS mutations significantly predicted worse overall survival with a hazard ratio of 2.01 (*p* < 0.001) (Table 4). Patients with liver metastasis with KRAS mutations had a significantly lower median overall survival of 30.2 months compared to 58.3 months in patients with liver metastasis without KRAS mutations (*p* = 0.00027) (Figure 1C). We did not observe a significant difference in OS with BRAF status among patients with liver metastasis (Appendix A). In the multivariate Cox regression analysis, KRAS mutations remained an independent predictor of overall survival in patients with liver metastasis (HR 2.29, *p* < 0.001), and notably, BRAF mutations were also noted to be independent predictors (HR 2.48, *p* = 0.027) (Table 4). Among patients with advanced CRC with non-liver metastasis, older age and BRAF mutations were found to be predictors of worse survival outcomes (HR 1.4, *p* = 0.0148 and HR 3.42, *p* = 0.006, respectively) (Table 4). Patients with non-liver metastasis with BRAF mutations had a median overall survival of 27.3 months, significantly less than the median overall survival of 49.5 months in patients without BRAF mutations (*p* = 0.0031) (Figure 1D). In multivariate Cox regression analysis, BRAF mutations remained a strong independent predictor of overall survival in patients with non-liver metastasis (HR 6.42, *p* = 0.001) (Table 4). Unlike the liver metastasis cohort, there was no association between KRAS mutations and survival outcomes among patients with non-liver metastasis.

We also examined the time from first-line to second-line treatment, defined as time from initiation of first-line treatment to initiation of second-line treatment, as a surrogate for treatment response. A cohort of 209 patients with known first- and second-line treatments was assessed (Appendix A). First-line therapy consisted of chemotherapy, molecular targeted therapy, and combinations of both, and treatment regimens were generally comparable between the liver metastasis and non-liver metastasis cohorts, which is fully described in Appendix A. Decision to initiate second-line therapy was based on treating physician discretion. Median time from first-line to second-line treatment was 14.3 months for all patients with advanced-stage CRC. Patients with liver metastasis had a median time from first-line to second-line treatment of 13 months, which was significantly shorter than the median time of 19.7 months in patients with non-liver metastasis (*p* = 0.00098) (Figure 1B). Multivariate Cox regression analysis identified the presence of liver metastasis as a significant predictor of shorter time on frontline therapy (HR 1.75, *p* < 0.001), suggesting systemic chemotherapy resistance in patients with liver metastasis. KRAS mutations were also found to be significant predictors of shorter time from first-line to second-line treatment, with an HR of 1.53 (*p* = 0.005) (Table 5). When examining factors impacting time from first-line to second-line treatment in patients with liver metastasis using Cox regression analysis, KRAS mutations were found to be a significant predictor of shorter time between treatments (HR 1.53, *p* = 0.012) (Appendix A). While patients with liver metastasis without KRAS mutations had a median time from first-line to second-line treatment of 14.1 months, patients with KRAS mutations had a shorter median time of 11.8 months (*p* = 0.012) (Appendix A). In multivariate Cox regression analysis, KRAS mutations continued to represent a significant independent predictor of shorter time between first-line and second-line treatments in patients with liver metastasis (HR 1.56, *p* = 0.012) (Appendix A). When examining patients with non-liver metastasis, no molecular alterations had a significant impact on time from first-line to second-line treatment (Appendix A).

## 4. Discussion

Liver metastasis of colorectal cancer is associated with worse survival outcomes and immunotherapy resistance, and the molecular underpinnings of this finding are not well defined. In our study, we did not identify any difference in the incidence of molecular alterations in KRAS, NRAS, BRAF, and other common oncogenic drivers and tumor suppressor genes among patients with advanced CRC with liver metastasis compared to those with non-liver metastasis, suggesting worse treatment response and inferior disease control with systemic therapy may not be related to distinct incidence of molecular alterations. Notably, we did not find a significant association between liver metastasis and overall survival outcomes, which is likely due to the small sample size, as well as patients with oligometastatic liver disease who underwent metastasectomy and were not captured in our study. Moreover, we identified that the presence of liver metastasis in patients with advanced CRC was associated with a shorter time from first-line to second-line treatment. Given that most patients in our study received chemotherapy as first-line treatment, these results suggest that the presence of liver metastasis is associated with inferior disease control on chemotherapy. Notably, when investigating the impact of individual molecular alterations on metastatic site-specific outcomes, we identified the predominant prognostic effect of BRAF mutation among patients with non-liver metastasis, while the effect of BRAF was less prominent among patients with liver metastasis. On the other hand, the presence of KRAS mutations was significantly predictive of worse outcomes in patients with liver metastasis but not in patients with non-liver metastasis.

The presence of liver metastasis in metastatic CRC is known to be associated with worse response to immunotherapy and worse survival outcomes regardless of MSI-H status [24,25]. In our study, we identified a shorter time from first-line to second-line therapy among patients with advanced-stage MSS CRC metastatic to the liver, suggesting liver involvement is not only associated with immunotherapy resistance, but potentially also with chemotherapy resistance. So far, there have been limited data in the literature that explore this biological and clinical difference seen among patients with liver metastasis of CRC. When investigating the mechanism behind this therapeutic resistance in patients with liver involvement, we did not identify any difference in the incidence of common molecular drivers of CRC, such as BRAF, KRAS, NRAS, and HER2, in patients with liver metastasis compared to those with non-liver metastasis. These findings indicate that therapeutic resistance seen in liver metastasis of CRC is likely related to the tumor microenvironment of the liver rather than clonal evolution of distinct molecular drivers. Consistent with our findings, recent landmark studies also showed that distinct molecular pathways and immune signatures impacting the tumor microenvironment may be associated with treatment resistance. A recent preclinical study of mouse models revealed that increased TGFβ activity in liver metastasis of CRC drives immunotherapy resistance by inducing T cell exclusion in the tumor microenvironment [26]. Another landmark preclinical study in which mouse models were utilized reported a macrophage-mediated immune exclusion process in the liver metastasis of CRC, leading to therapeutic resistance [27]. Moreover, there has been growing evidence from recent clinical trials highlighting the clinical relevance of the tumor microenvironment, including distant versus primary site of disease, which may create distinct outcomes. For example, although advanced-stage MSS CRC is known to be an immunologically cold tumor, highly promising responses were reported in the NICHE and NEST-1/NEST-2 trials among patients with localized MSS CRC, supporting our findings that the tumor microenvironment may have a direct impact on the therapeutic efficacy of systemic treatments [28,29,30,31,32].

In our study, we also identified distinct impact of molecular alterations in different metastatic sites of CRC, despite a similar incidence of such molecular alterations at each site. Most notably, we identified BRAF mutations as the leading prognostic factor for overall survival among patients with non-liver metastasis, while their impact among patients with liver metastasis, which represent the vast majority of our cohort, was borderline. Although this finding is highly novel, it should be interpreted within the context of a limited number of patients with BRAF mutations in the non-liver cohort (*n* = 7). It is also important to note that, while multivariate Cox regression analysis identified BRAF status as a significant independent predictor of overall survival among patients with liver metastasis, univariate analysis did not. This discrepancy is likely due to small sample size in the univariate analysis and control of confounding variables in multivariate analysis, suggesting that the prognostic effect of BRAF in the liver metastasis cohort is influenced by other variables such as TP53 status in the model. On the contrary, we found that the presence of KRAS mutations was predictive of both poor response to therapy and worse outcomes among patients with liver metastasis, but not among patients with non-liver metastasis. These findings suggest that, although the molecular characteristics of liver and non-liver metastasis are relatively similar, the impact of individual molecular alterations on the clinical course of patients with metastases at varying sites may be distinct, resulting in noticeably different survival outcomes. Such difference in impact of individual molecular alterations may also be related to differing tumor microenvironments of metastasis sites. These findings suggest that patients with KRAS-mutant liver metastasis may represent a population that could particularly benefit from effective KRAS-targeted therapies, which may reverse the poor survival outcomes and treatment resistance that we observe. Prospective clinical trials targeting the RAS oncogene should consider the biological and clinical relevance of the metastatic site-specific impact of oncogenic RAS mutations.

Our study has several limitations, including the retrospective nature of the study, which restricted our ability to capture potential confounding factors, the absence of the transcriptomic and proteomic portions of molecular data, the potential confounding impact of the variability of factors originating from clinician-driven decisions to switch therapies, and the limited size of the cohort with data on overall survival outcomes. The major strength of our study includes the novelty of scientific questions and findings as well as the collection of detailed clinical and molecular information which allowed us to perform extensive analyses to explore the molecular underpinnings of therapeutic resistance and differential clinical outcomes in patients with liver metastasis compared to those with non-liver metastasis, a topic that had not been well understood previously. Prospective studies with a larger cohort size are warranted to validate our findings.

## 5. Conclusions

Collectively, our study showed that the presence of liver metastasis in advanced CRC is associated with shorter time from first-line to second-line therapy, which may indicate resistance to chemotherapy. Our findings also suggest that such inferior disease control on systemic therapy among patients with liver metastasis is not linked to differing driver molecular alterations seen in CRC. Furthermore, despite similar incidence of molecular alterations in liver and non-liver metastases, driver alterations including BRAF and KRAS mutations may have a distinct impact on survival outcomes depending on the site of metastasis. These findings align with growing preclinical and clinical evidence that suggests the tumor microenvironment of liver metastasis may have highly distinct biological characteristics, resulting in an abrogated immune response and thus poorer response to systemic therapies, including chemotherapy and immunotherapy.

## Figures and Tables

**Figure 1 cancers-17-03677-f001:**
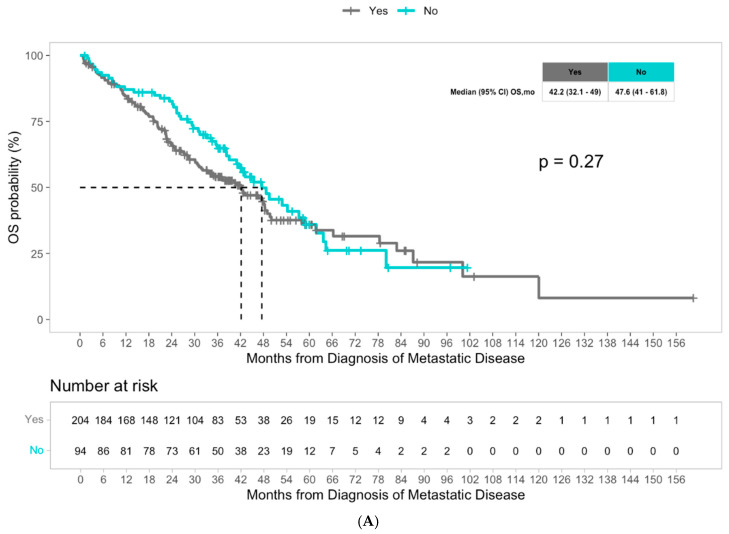
(**A**) Median overall survival in patients with advanced CRC with liver metastasis versus non-liver metastasis. (**B**) Median time from first-line to second-line treatment in patients with advanced CRC with liver metastasis versus non-liver metastasis. (**C**) Median overall survival in patients with advanced CRC with liver metastasis with KRAS mutations versus without KRAS mutations. (**D**) Overall median survival in patients with advanced CRC without liver metastasis with BRAF mutations versus without BRAF mutations.

**Table 1 cancers-17-03677-t001:** Demographic and clinical characteristics of overall cohort of patients with advanced CRC.

Characteristic		Patients with Advanced CRC (*n* = 299)
Age: mean [SD]		62.9 [11.3]
Gender	Female	126 (42%)
	Male	173 (58%)
Primary Site	Colon	233 (78%)
	Rectum	64 (21%)
	Unknown	2 (0.7%)
Liver Metastases	Yes	205 (69%)
	No	94 (31%)

**Table 2 cancers-17-03677-t002:** Molecular alterations and clinical characteristics of patients with advanced CRC with liver metastasis versus patients with advanced CRC with non-liver metastasis.

Molecular Alteration			Liver Metastasis (*n* = 205)	Non-Liver Metastasis (*n* = 94)	Overall (*n* = 299)	*p*-Value
KRAS	Overall		96 (47%)	51 (54%)	147 (49%)	0.263
	Exon 2 mutation		83/96 (86%)	44/51 (86%)	127/147 (86%)	1
	Non-exon 2 mutation		13/96 (14%)	7/51 (14%)	20/147 (14%)	1
NRAS			7 (3.4%)	4. (4.3%)	11 (3.7%)	0.746
BRAF			13 (6.3%)	7 (7.4%)	20 (6.7%)	0.804
TP53			170 (83%)	71 (76%)	241 (81%)	0.156
BRCA2			10 (4.9%)	4 (4.3%)	14 (4.7%)	1
PIK3CA			29 (14%)	15 (16%)	44 (15%)	0.726
ERRB2			2 (1.0%)	1 (1.1%)	3 (1.0%)	1
TMB (muts/Mb)			8.86	8.86	8.86	0.834
TMB (>10 muts/Mb)			80 (39%)	35 (37%)	115 (38%)	0.798
Gender		Male (*n* = 173)	121 (70%)	52 (30%)	173 (100%)	0.614
		Female (*n* = 126)	84 (67%)	42 (33%)	126 (100%)	
Primary tumor site		Colon (*n* = 233)	173 (74%)	60 (26%)	233 (100%)	0.00013
		Rectum (*n* = 64)	31 (48%)	33 (52%)	64 (100%)	

**Table 3 cancers-17-03677-t003:** Results of multivariate Cox regression analysis assessing impact of liver metastasis and molecular alterations on overall survival in patients with advanced CRC.

Variable	Multivariate Cox Regression Analysis	
	HR	*p*-Value
Liver metastases	1.28 [0.91, 1.79]	0.2
KRAS mutation	2.12 [1.50, 2.99]	<0.001
NRAS mutation	1.37 [0.61, 3.04]	0.4
BRAF mutation	3.37 [1.81, 6.25]	<0.001
TP53 mutation	1.24 [0.83, 1.86]	0.3
PIK3CA mutation	0.88 [0.56, 1.40]	0.6
BRCA2 mutation	1.13 [0.52, 2.44]	0.8

**Table 4 cancers-17-03677-t004:** Cox regression analysis, multivariate Cox regression analyses, and Kaplan–Meier survival analysis assessing the impact of age, CRC location, and molecular alterations on overall survival in patients with advanced CRC with liver metastasis and non-liver metastasis.

Liver					
Variable	Cox Regression Analysis		Multivariate Cox Regression Analysis		Kaplan–Meier Survival Analysis
	HR	*p*-Value	HR	*p*-Value	*p*-Value
Age at analysis	1.01 [0.99, 1.03]	0.3	N/A	N/A	N/A
Site of primary tumor (rectum/colon)	1.36 [0.81, 2.28]	0.3	N/A	N/A	0.25
KRAS	2.01 [1.37, 2.95]	<0.001	2.29 [1.52, 3.43]	<0.001	0.00027
NRAS	1.10 [0.43, 2.79]	0.8	1.43 [0.54, 3.81]	0.5	0.84
BRAF	1.59 [0.74, 3.43]	0.2	2.48 [1.11, 5.54]	0.027	0.23
TP53	1.23 [0.73, 2.07]	0.4	1.49 [0.87, 2.55]	0.15	0.43
BRCA2	1.03 [0.42, 2.54]	>0.9	0.94 [0.38, 2.35]	0.9	0.94
PIK3CA	1.06 [0.62, 1.84]	0.8	0.95 [0.54, 1.67]	0.9	0.82
TMB (categorical)	1.00 [0.68, 1.46]	>0.9	N/A	N/A	1
TMB (continuous)	1.00 [0.97, 1.03]	>0.9	N/A	N/A	N/A
**Non-Liver Metastasis Cohort**					
**Variable**	**Cox Regression Analysis**		**Multivariate Cox Regression Analysis**		**Kaplan–Meier Survival Analysis**
	**HR**	***p*-Value**	**HR**	***p*-Value**	***p*-Value**
Age at analysis	1.40 [1.07, 1.85]	0.0148	N/A	N/A	N/A
Site of primary tumor (rectum/colon)	0.59 [0.32, 1.08]	0.087	N/A	N/A	0.084
KRAS	1.31 [0.75, 2.30]	0.3	1.89 [0.95, 3.77]	0.069	0.35
NRAS	0.78 [0.19, 3.27]	0.7	1.16 [0.26, 5.20]	0.8	0.74
BRAF	3.42 [1.44, 8.16]	0.006	6.42 [2.12, 19.5]	0.001	0.0031
TP53	0.73 [0.40, 1.33]	0.3	0.87 [0.45, 1.69]	0.7	0.3
BRCA2	1.76 [0.42, 7.38]	0.4	2.65 [0.60, 11.70]	0.2	0.43
PIK3CA	0.95 [0.44, 2.02]	0.9	0.59 [0.25, 1.38]	0.2	0.89
TMB (categorical)	0.68 [0.37, 1.24]	0.2	N/A	N/A	0.2
TMB (continuous)	0.96 [0.92, 1.01]	0.1	N/A	N/A	N/A

**Table 5 cancers-17-03677-t005:** Results of multivariate Cox regression analysis assessing impact of liver metastasis and molecular alterations on time from first-line to second-line treatment in patients with advanced CRC.

Variable	Multivariate Cox Regression Analysis	
	HR	*p*-Value
Liver metastasis	1.75	<0.001
KRAS mutation	1.53	0.005
NRAS mutation	1.02	>0.9
BRAF mutation	1.45	0.2
TP53 mutation	0.97	0.9
PIK3CA mutation	1.08	0.7
BRCA2 mutation	1.10	0.8

## Data Availability

The data that support the findings of this study are available from the corresponding author upon reasonable request.

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
