# Peer review of "A Comparative Study of Clinical and Molecular Features of Microsatellite Stable Colorectal Cancer With and Without Liver Metastases"

_cancers, 2025, doi:10.3390/cancers17223677_

Round 1
Reviewer 1 Report
Comments and Suggestions for Authors
The authors conducted "A Comparative Study of Clinical and Molecular Features of Microsatellite Stable Colorectal Cancer with and without Liver Metastases" on cancers. The study addresses an important clinical question regarding therapeutic resistance in the context of liver metastases, particularly in microsatellite stable (MSS) CRC, where immunotherapy has limited activity. Generally, the manuscript is well written, but there are areas to be improved.
Major comments:
- Overall Survival (OS):
- While the paper states in the results and discussion that the difference in OS between liver and non-liver metastasis cohorts was "numerically worse" but "not statistically significant" (p=0.27), the Abstract and Simple Summary might give a slightly different meaning by initially stating, “Worse survival outcomes with liver metastasis of CRC are not well-defined. Please ensure that the abstract and simple summary clearly and consistently reflect the findings.
- Further, the abstract states that the "Despite the known association of liver metastasis with worse overall survival outcomes in broader literature, in our cohort, we observed numerically, but not statistically, worse median overall survival among patients with liver metastasis compared to those without liver metastasis (42.2 vs 47.6 months, p=0.27)." This is confusing.
- Novelty of site-specific prognostic impact:
- The finding that KRAS mutations are significant for OS in the liver metastasis cohort, while BRAF mutations are significant for OS in the non-liver metastasis cohort, is a highly significant and novel observation. This should be explicitly highlighted as a key discovery in the Abstract and Discussion. Consider rephrasing to emphasize its potential implications for precision medicine and treatment stratification.
- "Time to Next Treatment" as a surrogate:
- The definition of "time to next treatment" (TFS) as a surrogate for "treatment response" (lines 185-186) is well-justified and crucial to your findings regarding chemotherapy resistance. Please ensure this is clearly established and consistently referred to. The term "frontline therapy" is consistent. Briefly clarify in the abstract or simple summary that this refers to the duration of the first-line systemic therapy.
Author Response
Reviewer 1 Comments:
The authors conducted "A Comparative Study of Clinical and Molecular Features of Microsatellite Stable Colorectal Cancer with and without Liver Metastases" on cancers. The study addresses an important clinical question regarding therapeutic resistance in the context of liver metastases, particularly in microsatellite stable (MSS) CRC, where immunotherapy has limited activity. Generally, the manuscript is well written, but there are areas to be improved.
Major comments:
- Overall Survival (OS):
- While the paper states in the results and discussion that the difference in OS between liver and non-liver metastasis cohorts was "numerically worse" but "not statistically significant" (p=0.27), the Abstract and Simple Summary might give a slightly different meaning by initially stating, “Worse survival outcomes with liver metastasis of CRC are not well-defined. Please ensure that the abstract and simple summary clearly and consistently reflect the findings.
- Further, the abstract states that the "Despite the known association of liver metastasis with worse overall survival outcomes in broader literature, in our cohort, we observed numerically, but not statistically, worse median overall survival among patients with liver metastasis compared to those without liver metastasis (42.2 vs 47.6 months, p=0.27)." This is confusing.
- Novelty of site-specific prognostic impact:
- The finding that KRAS mutations are significant for OS in the liver metastasis cohort, while BRAF mutations are significant for OS in the non-liver metastasis cohort, is a highly significant and novel observation. This should be explicitly highlighted as a key discovery in the Abstract and Discussion. Consider rephrasing to emphasize its potential implications for precision medicine and treatment stratification.
- "Time to Next Treatment" as a surrogate:
- The definition of "time to next treatment" (TFS) as a surrogate for "treatment response" (lines 185-186) is well-justified and crucial to your findings regarding chemotherapy resistance. Please ensure this is clearly established and consistently referred to. The term "frontline therapy" is consistent. Briefly clarify in the abstract or simple summary that this refers to the duration of the first-line systemic therapy.
First, we would like to truly thank to the Reviewer for his/her time to review our article. Please find our point-by-point response below, addressing all the points of the Reviewer.
Q1) The Reviewer stated that “While the paper states in the results and discussion that the difference in OS between liver and non-liver metastasis cohorts was "numerically worse" but "not statistically significant" (p=0.27), the Abstract and Simple Summary might give a slightly different meaning by initially stating, “Worse survival outcomes with liver metastasis of CRC are not well-defined. Please ensure that the abstract and simple summary clearly and consistently reflect the findings. Further, the abstract states that the "Despite the known association of liver metastasis with worse overall survival outcomes in broader literature, in our cohort, we observed numerically, but not statistically, worse median overall survival among patients with liver metastasis compared to those without liver metastasis (42.2 vs 47.6 months, p=0.27)." This is confusing.”
A1) Thank you for pointing out the discrepancy. We have corrected the results section of the abstract to read “Although statistically not significant, we observed worse median overall survival among patients with liver metastasis (42.2 vs 47.6 months, p=0.27)” which is now consistent with both the simple summary and the paper’s findings. Now all three sections consistently state that we observed worse median overall survival among patients with liver metastasis, although not statistically significant, which is in line with the literature. Also, to clarify, in our abstract and paper, we did not state that worse survival outcomes with liver metastasis are not well-defined, we stated that the molecular and clinical underpinnings of worse overall survival are not well-defined.
Q2) The Reviewer stated that “The finding that KRAS mutations are significant for OS in the liver metastasis cohort, while BRAF mutations are significant for OS in the non-liver metastasis cohort, is a highly significant and novel observation. This should be explicitly highlighted as a key discovery in the Abstract and Discussion. Consider rephrasing to emphasize its potential implications for precision medicine and treatment stratification.”
A2) Thank you for this helpful suggestion. To further emphasize this point in the discussion section, we have added a sentence: “Such difference in impact of individual molecular alterations may also be related to differing tumor microenvironments of metastasis sites. These findings suggest patients with KRAS-mutant liver metastasis may especially benefit from KRAS-targeted therapies, which may reverse the poor survival outcomes and treatment resistance that we observe” We have also highlighted this point in the abstract: “Most notably, this is the first study to reveal that, despite similar incidence of molecular alterations, the impact of driver alterations including BRAF and KRAS mutations on survival outcomes can vary depending on the site of metastasis.”
Q3) The Reviewer stated that “The definition of "time to next treatment" (TFS) as a surrogate for "treatment response" (lines 185-186) is well-justified and crucial to your findings regarding chemotherapy resistance. Please ensure this is clearly established and consistently referred to. The term "frontline therapy" is consistent. Briefly clarify in the abstract or simple summary that this refers to the duration of the first-line systemic therapy.”
A3) Thank you for the comment. We have clarified in both the simple summary and abstract that time on frontline therapy was used as a surrogate for treatment response, thereby strengthening our findings regarding chemotherapy resistance.
Reviewer 2 Report
Comments and Suggestions for Authors
The authors present a comparative analysis of clinical and molecular features in patients with microsatellite stable (MSS) metastatic colorectal cancer (mCRC) with and without liver metastases. The study addresses a critical question in the field: why liver metastases are associated with poorer outcomes and treatment resistance. The key findings that liver metastases are linked to shorter time on frontline therapy (suggesting chemotherapy resistance) and that the prognostic impact of common driver mutations (KRAS, BRAF) is site-specific are novel and clinically significant. The manuscript is generally clear, the statistical analyses are appropriate, and the discussion effectively contextualizes the findings within the existing literature on the tumor microenvironment (TME). The work provides a solid foundation for future mechanistic and clinical studies. The authors have identified a clinically relevant pattern of accelerated treatment failure and a metastasis site-specific role for driver mutations. Addressing the major comments, particularly regarding the definition of treatment progression and the nuance of the "resistance" claim, will significantly strengthen the manuscript and its impact.
Major Comments
The finding that liver metastasis predicts a shorter "time to next treatment" (TNT) is a cornerstone of the manuscript. However, the criteria for initiating second-line therapy are not defined. Were these decisions based on radiographic progression (RECIST criteria), clinical deterioration, or physician discretion? Clarifying this is crucial, as the interpretation of TNT as a surrogate for "chemotherapy resistance" hinges on the assumption that progression was the primary reason for switching therapy.
Relatedly, Table S2 (which details first-line therapies) is referenced but not included in the provided text. It is essential to include a summary of this data in the main manuscript or supplementary materials to demonstrate that the treatment regimens were comparable between the liver metastasis and non-liver metastasis cohorts. A significant imbalance in, for example, the use of biologics (anti-EGFR vs. anti-VEGF) could be a major confounder.
The conclusion that liver metastases confer "chemotherapy resistance" is strong and should be nuanced. The data robustly shows that disease progresses faster to the point of requiring second-line therapy. However, the term "resistance" often implies a lack of initial response. The data presented cannot distinguish between: A. Intrinsic resistance (no initial shrinkage); B. Acquired resistance (initial response but rapid progression); C. A more aggressive disease biology leading to faster progression in new sites, not necessarily resistance in the treated lesions.
The authors should temper their language accordingly (e.g., "shorter time on frontline therapy," "inferior disease control," "shorter treatment benefit") unless they have data on objective response rates (ORR) or depth of response (DpR) to first-line therapy that could support the specific mechanism of resistance.
The lack of a statistically significant difference in overall survival (OS) between the liver and non-liver metastasis cohorts (42.2 vs. 47.6 months, p=0.27) is noted, but the potential reasons (sample size, metastasectomy) are appropriately discussed. A power calculation or a comment on the observed effect size would strengthen this discussion. Was the study powered to detect a specific hazard ratio for OS?
In the multivariate Cox models for site-specific cohorts (Table 4), some subgroups are very small (e.g., only 7 patients with BRAF mutations in the non-liver metastasis cohort). The large Hazard Ratios (e.g., HR=6.42 for BRAF in non-liver mets), while likely real, should be interpreted with caution due to the wide confidence intervals that such small numbers produce. Reporting the 95% Confidence Intervals for these key HRs in the tables would be highly valuable for the reader.
Minor Comments
The Simple Summary states: "Our study also reveals that molecular features of liver metastases of CRC are relatively similar to those without liver metastases..." This is slightly contradicted by the central finding of the differential impact of KRAS/BRAF. Consider rephrasing to emphasize that while the incidence of mutations is similar, their prognostic impact is site-specific.
In the Abstract Results, the sentence "Although statistically not significant, we observed worse median overall survival among patients with non-liver metastasis (42.2 vs 47.6 months, p=0.27)" is confusing. It reads as if the non-liver metastasis group had worse survival. It should be rephrased to: "...worse median overall survival among patients with liver metastasis (42.2 months) compared to those with non-liver metastasis (47.6 months, p=0.27)."
Page 4, Table 2: The p-value for "Primary tumor site" is listed twice as 0.00013. It would be clearer to place a single p-value in the header for that section or label one as "Colon" and the other as "Rectum" for clarity.
Page 5, Lines 163-164: It is stated that in the liver metastasis cohort, "BRAF mutations were also noted to be independent predictors (HR 2.48, p = 0.027)" in the multivariate analysis. However, this seems to contrast with the earlier statement on lines 160-161 ("We did not observe a significant difference in OS with BRAF status...") and the Kaplan-Meier p-value of 0.23 in Table 4. This discrepancy between the univariate (KM) and multivariate analyses should be briefly discussed. It suggests that the prognostic effect of BRAF in the liver mets cohort is influenced by other variables in the model.
Referencing Figure S1 and S2 in the main text is good practice, and these should be included in the final submission.
Author Response
Reviewer 2 Comments:
The authors present a comparative analysis of clinical and molecular features in patients with microsatellite stable (MSS) metastatic colorectal cancer (mCRC) with and without liver metastases. The study addresses a critical question in the field: why liver metastases are associated with poorer outcomes and treatment resistance. The key findings that liver metastases are linked to shorter time on frontline therapy (suggesting chemotherapy resistance) and that the prognostic impact of common driver mutations (KRAS, BRAF) is site-specific are novel and clinically significant. The manuscript is generally clear, the statistical analyses are appropriate, and the discussion effectively contextualizes the findings within the existing literature on the tumor microenvironment (TME). The work provides a solid foundation for future mechanistic and clinical studies. The authors have identified a clinically relevant pattern of accelerated treatment failure and a metastasis site-specific role for driver mutations. Addressing the major comments, particularly regarding the definition of treatment progression and the nuance of the "resistance" claim, will significantly strengthen the manuscript and its impact.
Major Comments
- The finding that liver metastasis predicts a shorter "time to next treatment" (TNT) is a cornerstone of the manuscript. However, the criteria for initiating second-line therapy are not defined. Were these decisions based on radiographic progression (RECIST criteria), clinical deterioration, or physician discretion? Clarifying this is crucial, as the interpretation of TNT as a surrogate for "chemotherapy resistance" hinges on the assumption that progression was the primary reason for switching therapy.
- Relatedly, Table S2 (which details first-line therapies) is referenced but not included in the provided text. It is essential to include a summary of this data in the main manuscript or supplementary materials to demonstrate that the treatment regimens were comparable between the liver metastasis and non-liver metastasis cohorts. A significant imbalance in, for example, the use of biologics (anti-EGFR vs. anti-VEGF) could be a major confounder.
- The conclusion that liver metastases confer "chemotherapy resistance" is strong and should be nuanced. The data robustly shows that disease progresses faster to the point of requiring second-line therapy. However, the term "resistance" often implies a lack of initial response. The data presented cannot distinguish between: A. Intrinsic resistance (no initial shrinkage); B. Acquired resistance (initial response but rapid progression); C. A more aggressive disease biology leading to faster progression in new sites, not necessarily resistance in the treated lesions. The authors should temper their language accordingly (e.g., "shorter time on frontline therapy," "inferior disease control," "shorter treatment benefit") unless they have data on objective response rates (ORR) or depth of response (DpR) to first-line therapy that could support the specific mechanism of resistance.
- The lack of a statistically significant difference in overall survival (OS) between the liver and non-liver metastasis cohorts (42.2 vs. 47.6 months, p=0.27) is noted, but the potential reasons (sample size, metastasectomy) are appropriately discussed. A power calculation or a comment on the observed effect size would strengthen this discussion. Was the study powered to detect a specific hazard ratio for OS?
- In the multivariate Cox models for site-specific cohorts (Table 4), some subgroups are very small (e.g., only 7 patients with BRAF mutations in the non-liver metastasis cohort). The large Hazard Ratios (e.g., HR=6.42 for BRAF in non-liver mets), while likely real, should be interpreted with caution due to the wide confidence intervals that such small numbers produce. Reporting the 95% Confidence Intervals for these key HRs in the tables would be highly valuable for the reader.
Minor Comments
- The Simple Summary states: "Our study also reveals that molecular features of liver metastases of CRC are relatively similar to those without liver metastases..." This is slightly contradicted by the central finding of the differential impact of KRAS/BRAF. Consider rephrasing to emphasize that while the incidence of mutations is similar, their prognostic impact is site-specific.
- In the Abstract Results, the sentence "Although statistically not significant, we observed worse median overall survival among patients with non-liver metastasis (42.2 vs 47.6 months, p=0.27)" is confusing. It reads as if the non-liver metastasis group had worse survival. It should be rephrased to: "...worse median overall survival among patients with liver metastasis (42.2 months) compared to those with non-liver metastasis (47.6 months, p=0.27)."
- Page 4, Table 2: The p-value for "Primary tumor site" is listed twice as 0.00013. It would be clearer to place a single p-value in the header for that section or label one as "Colon" and the other as "Rectum" for clarity.
- Page 5, Lines 163-164: It is stated that in the liver metastasis cohort, "BRAF mutations were also noted to be independent predictors (HR 2.48, p = 0.027)" in the multivariate analysis. However, this seems to contrast with the earlier statement on lines 160-161 ("We did not observe a significant difference in OS with BRAF status...") and the Kaplan-Meier p-value of 0.23 in Table 4. This discrepancy between the univariate (KM) and multivariate analyses should be briefly discussed. It suggests that the prognostic effect of BRAF in the liver mets cohort is influenced by other variables in the model.
- Referencing Figure S1 and S2 in the main text is good practice, and these should be included in the final submission.
First, we would like to thank the Reviewer for their very thoughtful comments, and here is our answer:
Q1) The reviewer pointed out “The finding that liver metastasis predicts a shorter "time to next treatment" (TNT) is a cornerstone of the manuscript. However, the criteria for initiating second-line therapy are not defined. Were these decisions based on radiographic progression (RECIST criteria), clinical deterioration, or physician discretion? Clarifying this is crucial, as the interpretation of TNT as a surrogate for "chemotherapy resistance" hinges on the assumption that progression was the primary reason for switching therapy.
A1) Thanks so much for drawing attention to this point. We have added a line in the results section to clarify this: “Decision to initiate second-line therapy was based on treating physician discretion.”
Q2) The reviewer also stated “Relatedly, Table S2 (which details first-line therapies) is referenced but not included in the provided text. It is essential to include a summary of this data in the main manuscript or supplementary materials to demonstrate that the treatment regimens were comparable between the liver metastasis and non-liver metastasis cohorts. A significant imbalance in, for example, the use of biologics (anti-EGFR vs. anti-VEGF) could be a major confounder.
A2) Thanks so much for pointing this out. The data on treatment regimens are included in a supplemental table. We also added a line in the results stating that treatment regimens were comparable between the liver metastasis and non-liver metastasis cohorts: “First-line therapy consisted of chemotherapy, molecular targeted therapy, and combinations of both, and treatment regimens were generally comparable between liver metastasis and non-liver metastasis cohorts, which is fully described in Table S2.”
Q3) The reviewer stated “The conclusion that liver metastases confer "chemotherapy resistance" is strong and should be nuanced. The data robustly shows that disease progresses faster to the point of requiring second-line therapy. However, the term "resistance" often implies a lack of initial response. The data presented cannot distinguish between: A. Intrinsic resistance (no initial shrinkage); B. Acquired resistance (initial response but rapid progression); C. A more aggressive disease biology leading to faster progression in new sites, not necessarily resistance in the treated lesions. The authors should temper their language accordingly (e.g., "shorter time on frontline therapy," "inferior disease control," "shorter treatment benefit") unless they have data on objective response rates (ORR) or depth of response (DpR) to first-line therapy that could support the specific mechanism of resistance.”
A3) Thanks so much for a very important and helpful point. We have consistently toned down our language surrounding chemotherapy resistance to be more nuanced throughout the Simple Summary, abstract, and paper. For instance, our Simple Summary now reads “In our study, we also identified that liver metastasis of CRC is associated with shorter time on frontline therapy, a surrogate for treatment response. This suggests that inferior treatment response seen with liver metastasis is not limited to immunotherapy but also applies to chemotherapy.” Our abstract reads: “Liver metastasis of CRC is associated with shorter time on frontline therapy, indicative of potential chemotherapy resistance.” Our discussion reads: “We identified that the presence of liver metastasis in patients with advanced CRC was associated with a shorter time from first-line to second-line treatment. Given that most patients in our study received chemotherapy as first-line treatment, these results suggest that the presence of liver metastasis is associated with inferior disease control on chemotherapy.” Our conclusion reads: “Collectively, our study showed that the presence of liver metastasis in advanced CRC is associated with shorter time from first-line to second-line therapy, which may indicate resistance to chemotherapy. Our findings also suggest that such inferior disease control on systemic therapy among patients with liver metastasis is not linked to differing driver molecular alterations seen in CRC.”
Q4) The reviewer asked “The lack of a statistically significant difference in overall survival (OS) between the liver and non-liver metastasis cohorts (42.2 vs. 47.6 months, p=0.27) is noted, but the potential reasons (sample size, metastasectomy) are appropriately discussed. A power calculation or a comment on the observed effect size would strengthen this discussion. Was the study powered to detect a specific hazard ratio for OS?”
A4) Thanks so much for this question. We did not do a pre-study power analysis due to the fact we have a pre-defined study population that was limited to the available cohort. Please also note that we did acknowledge the size of cohort as one of the limitations of our study.
Q5) The reviewer noted, “In the multivariate Cox models for site-specific cohorts (Table 4), some subgroups are very small (e.g., only 7 patients with BRAF mutations in the non-liver metastasis cohort). The large Hazard Ratios (e.g., HR=6.42 for BRAF in non-liver mets), while likely real, should be interpreted with caution due to the wide confidence intervals that such small numbers produce. Reporting the 95% Confidence Intervals for these key HRs in the tables would be highly valuable for the reader.”
A5) Thanks so much for this suggestion. We have reported the 95% confidence intervals for HRs in the tables accordingly throughout the manuscript.
Q6) The reviewer suggested, “The Simple Summary states: "Our study also reveals that molecular features of liver metastases of CRC are relatively similar to those without liver metastases..." This is slightly contradicted by the central finding of the differential impact of KRAS/BRAF. Consider rephrasing to emphasize that while the incidence of mutations is similar, their prognostic impact is site-specific.”
A6) Thanks so much for pointing this out. We have rephrased this in the Simple Summary, abstract, and conclusion section. The Simple Summary now reads: “Our study also revealed that the incidence of molecular alterations is relatively similar between liver metastases and non-liver metastases of CRC…. We also discovered that the metastatic site-specific impact of driver oncogenes, such as BRAFS and KRAS mutations, pointing out that the impact of driver alterations on survival outcomes can vary depending on the site of metastasis of CRC.” The abstract now reads: “Despite similar incidence of molecular alterations, driver alterations including BRAF and KRAS mutations may have a distinct impact on survival outcomes depending on the site of metastasis.” The conclusion now reads: “Furthermore, despite similar incidence of molecular alterations in liver and non-liver metastases, driver alterations including BRAF and KRAS mutations may have a distinct impact on survival outcomes depending on the site of metastasis.”
Q7) The reviewer pointed out an error: “In the Abstract Results, the sentence "Although statistically not significant, we observed worse median overall survival among patients with non-liver metastasis (42.2 vs 47.6 months, p=0.27)" is confusing. It reads as if the non-liver metastasis group had worse survival. It should be rephrased to: "...worse median overall survival among patients with liver metastasis (42.2 months) compared to those with non-liver metastasis (47.6 months, p=0.27)."
A7) Thanks so much for pointing this out. This has been corrected to read “Although not statistically not significant, we observed worse median overall survival among patients with liver metastasis (42.2 vs 47.6 months, p=0.27).”
Q8) The reviewer pointed out that “Page 4, Table 2: The p-value for "Primary tumor site" is listed twice as 0.00013. It would be clearer to place a single p-value in the header for that section or label one as "Colon" and the other as "Rectum" for clarity.
A8) Thanks for this comment. We are unsure what is meant by this comment. We do not see the p-value listed twice as 0.00013. Instead, we only see one p-value listed for primary tumor site in Table 2 on Page 4. This p-value is shared between both the colon and rectum primary sites, the same way that both male and female gender share a p-value. We have ensured there is no duplication of this p-value anywhere in the paper.
Q9) The reviewer stated “Page 5, Lines 163-164: It is stated that in the liver metastasis cohort, "BRAF mutations were also noted to be independent predictors (HR 2.48, p = 0.027)" in the multivariate analysis. However, this seems to contrast with the earlier statement on lines 160-161 ("We did not observe a significant difference in OS with BRAF status...") and the Kaplan-Meier p-value of 0.23 in Table 4. This discrepancy between the univariate (KM) and multivariate analyses should be briefly discussed. It suggests that the prognostic effect of BRAF in the liver mets cohort is influenced by other variables in the model.”
A9) Thanks so much for this suggestion. We have discussed the discrepancy between the univariate and multivariate analyses in the paper discussion: “Specifically, while multivariate Cox regression analysis identified BRAF status as a significant independent predictor of overall survival among patients with liver metastasis, univariate analysis did not. This discrepancy is likely due to small sample size in the univariate analysis and control of confounding variables in multivariate analysis, suggesting that the prognostic effect of BRAF in the liver metastasis cohort is influenced by other variables such as TP53 status in the model.”
Q10) The reviewer suggested, “Referencing Figure S1 and S2 in the main text is good practice, and these should be included in the final submission.”
A10) Thanks so much for pointing this out. Figure S1 and S2 are referenced in the main text. We will also be sure to ensure that both figures are uploaded and included in the final submission.
Reviewer 3 Report
Comments and Suggestions for Authors
This is a valuable study that provides compelling evidence for liver metastasis as a marker of broader systemic therapy resistance in MSS mCRC. The site-specific prognostic role of KRAS and BRAF mutations is a significant finding.
1.Perform and report an analysis of the mean of the number of metastatic liver lesion.
2. Correct the typo "BRAS" to "BRAF" in the Simple Summary. Harmonize the language in the Simple Summary with the Abstract .
3.Briefly discuss the discrepancy between the non-significant univariate and significant multivariate results for BRAF in the liver metastasis cohort.
4.Briefly mention the potential clinical implications of the findings (e.g., whether patients with KRAS-mutant liver metastases warrant a different treatment approach).
Author Response
Reviewer 3 Comments:
This is a valuable study that provides compelling evidence for liver metastasis as a marker of broader systemic therapy resistance in MSS mCRC. The site-specific prognostic role of KRAS and BRAF mutations is a significant finding.
- Perform and report an analysis of the mean of the number of metastatic liver lesion.
- Correct the typo "BRAS" to "BRAF" in the Simple Summary. Harmonize the language in the Simple Summary with the Abstract .
- Briefly discuss the discrepancy between the non-significant univariate and significant multivariate results for BRAF in the liver metastasis cohort.
- Briefly mention the potential clinical implications of the findings (e.g., whether patients with KRAS-mutant liver metastases warrant a different treatment approach).
First, we would like to thank the Reviewer for his/her helpful and thoughtful comments. Below is our response:
Q1) The reviewer suggested performing and reporting an analysis of the mean number of metastatic liver lesions.
A1) We appreciate this suggestion. Unfortunately, our database lacks information on the exact number and size of metastases in each organ, so this is not a feasible change. This question would be best addressed in prospective studies, although it should be noted that there is rarely this level of detail in most prospective studies and clinical trials. Nonetheless, this represents an important scientific question and unfortunately our database is lacking relevant information to answer this question.
Q2) The reviewer recommended correcting the typo "BRAS" to "BRAF" in the Simple Summary and harmonizing the language in the Simple Summary with the Abstract.
A2) Thanks so much for pointing this out. The typo has been corrected in the Simple Summary. The language in the Simple Summary and the Abstract has also been harmonized to read more similarly.
Q3) The reviewer suggested we briefly discuss the discrepancy between the non-significant univariate and significant multivariate results for BRAF in the liver metastasis cohort.
A3) Thanks so much for this suggestion. We commented on this discrepancy in the paper discussion, stating “Specifically, while multivariate Cox regression analysis identified BRAF status as a significant independent predictor of overall survival among patients with liver metastasis, univariate analysis did not. This discrepancy is likely due to small sample size in the univariate analysis and control of confounding variables in multivariate analysis, suggesting that the prognostic effect of BRAF in the liver metastasis cohort is influenced by other variables such as TP53 status in the model.”
Q4) The reviewer suggested that we briefly mention the potential clinical implications of the findings (e.g., whether patients with KRAS-mutant liver metastases warrant a different treatment approach).
A4) Thanks so much for this suggestion. We added clinical implications to the paper discussion: “These findings suggest patients with KRAS-mutant liver metastasis may potentially benefit from KRAS-targeted therapies, which may reverse the poor survival outcomes and treatment resistance that we observe.”
Round 2
Reviewer 1 Report
Comments and Suggestions for Authors
The revised manuscript is stronger, and the authors have done an excellent job of presenting their data and arguments. However, further revisions are required.
Major comments
- Interpretation of "Time on Frontline Therapy" as a Surrogate for Chemoresistance:
The use of "time from first line to second-line treatment" as a surrogate for treatment response is a central pillar of the study's conclusions. While this is a reasonable and commonly used real-world endpoint, its interpretation as a direct measure of "chemotherapy resistance" requires careful framing. The decision to switch therapies is subject to various clinical factors beyond objective radiographic progression, including physician judgment, patient performance status, and symptom burden. Patients with liver metastases may be perceived as having a poorer prognosis, potentially leading clinicians to switch treatments at a lower threshold.
In the Discussion (e.g., around lines 439-442), please explicitly acknowledge this limitation. A brief statement clarifying that while this endpoint is strongly suggestive of poorer disease control, it can be influenced by clinical factors other than objective resistance would add important nuance. The current phrasing in the conclusion ("which may indicate resistance to chemotherapy") is good, but strengthening the discussion of this point would be beneficial.
- Statistical Power and Interpretation of Site-Specific Prognostic Biomarkers:
The finding regarding the site-specific prognostic impact of BRAF and KRAS mutations is the most novel and exciting aspect of this paper. However, this conclusion should be tempered by the limited statistical power in the subgroup analyses. Specifically, in the non-liver metastasis cohort (N=94), the powerful prognostic effect of BRAF mutation (HR 6.42, p=0.001 in multivariate analysis) is derived from a very small number of patients (n=7, per Table 2). The extremely wide 95% confidence interval [2.12, 19.51] reflects this imprecision.
While the finding is significant and should be reported, the authors should be more explicit about this limitation in the discussion. Please add a sentence cautioning that the magnitude of the effect of BRAF in the non-liver metastasis cohort should be interpreted with care due to the small number of events and that this finding requires validation in larger, independent cohorts. This will prevent overinterpretation of this otherwise compelling result.
Minor comments
- Consistency of terminology: There appears to be a minor inconsistency in the terminology used for the primary treatment response endpoint. The Y-axis in Figure 1B is labelled "TFS probability (%)" (Treatment-Free Survival), whereas the text and figure caption refer to "Median time from first line to second-line treatment." While related, these are distinct metrics.
Please ensure the terminology is consistent throughout the manuscript. "Time-to-next-treatment" (TTNT) or "time to second-line treatment" might be more precise than TFS. Please update the figure axis label to match the text for clarity.
- Abstract: The revised summary and abstract are clear. However, a few phrases could be refined for precision.
- Line 66: The phrase "inferior treatment response, seen with liver metastasis, is not limited to immunotherapy but also may apply to chemotherapy" is excellent.
- Line 145: The phrase "Although not statistically significant, we observed worse median overall survival" is slightly awkward. A more direct phrasing would be: "There was a trend toward worse median overall survival among patients with liver metastasis, though this was not statistically significant (42.2 vs 47.6 months, p=0.27)."
- Specificity of "Other Molecular Alterations": In the Results section (page 4, lines 254-255), the text states: "There was no significant difference in the incidence of KRAS, NRAS, BRAF, or other molecular alterations..." The term "other" is slightly vague. To improve specificity, please consider rephrasing to "...or other molecular alterations tested, including TP53, BRCA2, and PIK3CA (Table 2)."
- Strength of Conclusion Regarding KRAS-Targeted Therapies: In the Discussion (line 555), the manuscript suggests that "patients with KRAS-mutant liver metastasis may especially benefit from KRAS-targeted therapies." This is a reasonable and forward-looking hypothesis. However, given that this study is based on prognostic (not predictive) data, a slightly more cautious tone may be warranted. Consider softening the language slightly, for example: "...suggests that patients with KRAS-mutant liver metastasis may represent a population that could particularly benefit from effective KRAS-targeted therapies..."
Author Response
Q1) The Reviewer stated that “Interpretation of "Time on Frontline Therapy" as a Surrogate for Chemoresistance:
The use of "time from first line to second-line treatment" as a surrogate for treatment response is a central pillar of the study's conclusions. While this is a reasonable and commonly used real-world endpoint, its interpretation as a direct measure of "chemotherapy resistance" requires careful framing. The decision to switch therapies is subject to various clinical factors beyond objective radiographic progression, including physician judgment, patient performance status, and symptom burden. Patients with liver metastases may be perceived as having a poorer prognosis, potentially leading clinicians to switch treatments at a lower threshold.
In the Discussion (e.g., around lines 439-442), please explicitly acknowledge this limitation. A brief statement clarifying that while this endpoint is strongly suggestive of poorer disease control, it can be influenced by clinical factors other than objective resistance would add important nuance. The current phrasing in the conclusion ("which may indicate resistance to chemotherapy") is good, but strengthening the discussion of this point would be beneficial.”
A1) Thank you for pointing out this limitation, and we fully agree with the reviewer. We have addressed this point in limitations by adding “the potential confounding impact of the variability of factors originating from clinicians-driven decisions to switch therapies. “
Q2) The Reviewer stated that “Statistical Power and Interpretation of Site-Specific Prognostic Biomarkers:
The finding regarding the site-specific prognostic impact of BRAF and KRAS mutations is the most novel and exciting aspect of this paper. However, this conclusion should be tempered by the limited statistical power in the subgroup analyses. Specifically, in the non-liver metastasis cohort (N=94), the powerful prognostic effect of BRAF mutation (HR 6.42, p=0.001 in multivariate analysis) is derived from a very small number of patients (n=7, per Table 2). The extremely wide 95% confidence interval [2.12, 19.51] reflects this imprecision.
While the finding is significant and should be reported, the authors should be more explicit about this limitation in the discussion. Please add a sentence cautioning that the magnitude of the effect of BRAF in the non-liver metastasis cohort should be interpreted with care due to the small number of events and that this finding requires validation in larger, independent cohorts. This will prevent overinterpretation of this otherwise compelling result.
”
A2) Thank you for this helpful suggestion. We completely agree with the Reviewer. To further emphasize this point in the discussion section, we have added a sentence: “Although this finding is highly novel, it should be interpreted within the context of a limited number of patients with BRAF mutations in the non-liver cohort (N=7).”
Q3) The Reviewer stated that “Consistency of terminology: There appears to be a minor inconsistency in the terminology used for the primary treatment response endpoint. The Y-axis in Figure 1B is labelled "TFS probability (%)" (Treatment-Free Survival), whereas the text and figure caption refer to "Median time from first line to second-line treatment." While related, these are distinct metrics.
Please ensure the terminology is consistent throughout the manuscript. "Time-to-next-treatment" (TTNT) or "time to second-line treatment" might be more precise than TFS. Please update the figure axis label to match the text for clarity.
A3) Thank you for the comment. We have now updated Figure 1B according to the input of the Reviewer for consistency.
Q4) The Reviewer stated that “Abstract: The revised summary and abstract are clear. However, a few phrases could be refined for precision.
- Line 66: The phrase "inferior treatment response, seen with liver metastasis, is not limited to immunotherapy but also may apply to chemotherapy" is excellent.
- Line 145: The phrase "Although not statistically significant, we observed worse median overall survival" is slightly awkward. A more direct phrasing would be: "There was a trend toward worse median overall survival among patients with liver metastasis, though this was not statistically significant (42.2 vs 47.6 months, p=0.27)."
A4) Thanks for this wise comment. We completely agree with the Reviewer and have now updated Line 145 with the sentence suggested by the reviewer: “There was a trend toward worse median overall survival among patients with liver metastasis, though this was not statistically significant (42.2 vs 47.6 months, p=0.27)”. We also appreciate his/her positive feedback on changes to Line 66.
Q5) The Reviewer stated that Specificity of "Other Molecular Alterations": In the Results section (page 4, lines 254-255), the text states: "There was no significant difference in the incidence of KRAS, NRAS, BRAF, or other molecular alterations..." The term "other" is slightly vague. To improve specificity, please consider rephrasing to "...or other molecular alterations tested, including TP53, BRCA2, and PIK3CA (Table 2)."
A5) Thanks for this meaningful comment as well. We completely agree with the Reviewer and address the point as follows: “There was no significant difference in the incidence of KRAS, NRAS, BRAF, or other molecular alterations tested, including TP53, BRCA2, and PIK3CA in patients with liver metastasis compared to patients with non-liver metastasis (Table 2).”
Q6) The Reviewer stated that “Strength of Conclusion Regarding KRAS-Targeted Therapies: In the Discussion (line 555), the manuscript suggests that 'patients with KRAS-mutant liver metastasis may especially benefit from KRAS-targeted therapies." This is a reasonable and forward-looking hypothesis. However, given that this study is based on prognostic (not predictive) data, a slightly more cautious tone may be warranted. Consider softening the language slightly, for example: "...suggests that patients with KRAS-mutant liver metastasis may represent a population that could particularly benefit from effective KRAS-targeted therapies..."
A6) Once again, we would like to applaud for this meaningful comment of the Reviewer. We fully agree and now addressed the sentences as: “These findings suggest that patients with KRAS-mutant liver metastasis may represent a population that could particularly benefit from effective KRAS-targeted therapies.”
Reviewer 2 Report
Comments and Suggestions for Authors
No further comments
Author Response
Thanks for endorsing our article.